# A2FC: A FEDERATED ADVANTAGE ACTOR-CRITIC LEARNING APPROACH FOR HETEROGENEOUS ACTION SPACES

## ABSTRACT

The growth of the Internet of Things (IoT) and the increasing demand for real-time networking have brought about a growing necessity for multiple reinforcement learning (RL) agents to collaboratively train within a shared environment, all working towards common objectives. The multi-agent Advantage Actor-Critic (A2C) algorithm is gaining popularity in Multi-Agent Reinforcement Learning (MARL) systems. However, this approach requires agents to share policy components among neighboring agents due to observations being only partially available to each agent. This practice increases communication overhead and raises privacy concerns. Federated learning (FL), recognized as a privacy-preserving machine learning method, can be applied in the MARL context with a central server aggregating the weights of the agents' actor and critic models. However, this technique assumes that all agents are capable of executing identical actions, which may be impractical. To overcome the aforementioned shortcomings, we introduce a novel FL A2C algorithm called "Advantage Actor Federated Critic (A2FC)". The proposed algorithm streamlines the aggregation of agents' critic models while offloading the training of actor models to the individual agents' local machines. An empirical experiment conducted in an adaptive traffic signal control (ATSC) system demonstrates the method's effectiveness in personalizing agents' actions, preserving agents' privacy during training, and mitigating communication overhead issues.

## 1 INTRODUCTION

RL is a systematic mathematical framework for self-guided learning driven by experiences, defined within the context of a Markov Decision Process (MDP). RL aims to train autonomous agents to acquire optimal behaviors in interactive environments. Deep reinforcement learning (DRL), first introduced in (Mnih et al., 2015), extends RL's capabilities by integrating deep learning models and RL techniques. RL can be categorized into value-based methods, policy-based methods, and actor-critic methods. *Q learning* (Watkins & Dayan, 1992), a prime example of a value-based method, uses a state-action value function that is updated incrementally based on experiences. However, such methods are "off-policy", updating policies using one-step *temporal differences*, which assumes stationary MDP transitions. In contrast, policy-based methods like *REINFORCE* (Williams, 1992) are "on-policy", updating policies using sampled episodic returns. Actor-critic methods (Konda & Tsitsiklis, 1999) combine aspects of both value-based and policy-based methods. They separate the policy function from the value function, thus reducing bias and variance compared to policy-based approaches. An advancement is A2C (Mnih et al., 2016), which introduces an estimated advantage value, indicating how superior an action is at the current state.

With the proliferation of IoT and smart devices, there is a growing demand for multiple RL agents to collaboratively train in shared environments to achieve global objectives. These environments can be characterized as cooperative MARL systems. For example, consider an adaptive traffic signal control (ATSC) system where multiple RL agents work together to dynamically adjust traffic signal timings, aiming to reduce traffic congestion across the entire area. Each RL agent is responsible for controlling only one signalized intersection. Despite the potential benefits, existing MARL algorithms face certain challenges. Traditional MARL algorithms, based on Q-learning, distribute the

global Q-function to local agents (Guestrin et al., 2002; Kok & Vlassis, 2006). These approaches rely on strict rules to balance scalability and optimality. In contrast, Independent Q-learning (Tan, 1993) enhances scalability as each local agent independently trains its policy, treating others as parts of the observable environment. However, Q-learning's reliance on tabular structures limits its scalability, particularly in real-world applications with resource-constrained devices. Moreover, tabular structures struggle to leverage previous experiences or shared knowledge in similar states.

The A2C algorithm is considered a simpler approach than Q-learning in a large-scale MARL system. RL agents apply the A2C algorithm within the multi-agent framework to achieve quicker and more straightforward convergence in MARL (Chu et al., 2019). The introduction of FL (Konečný et al., 2016a;b; McMahan et al., 2016), a concept that has gained considerable attention in the field of privacy-preserving machine learning, has expanded the potential applications of algorithms like A2C, which rely on neural networks. FL involves the coordination of multiple clients, including computers, processing devices, or smart sensors, by a central server to collectively train a machine learning model. There have been efforts to combine FL with DRL to enhance its robustness to similar but previously unseen states (Wang et al., 2020a; Ye et al., 2021). However, these approaches also raise concerns about the required communication overhead and privacy leakage. From the perspective of communication, agents are required to send and receive others' policies as parts of inputs in the multi-agent setting, or send and receive neural networks parameters of both actor and critic in the FL setting, which creates a heavy communication load, especially when agents need to make rapid responses. Meanwhile, privacy leakage is the second concern when multiple agents are required to share their trained policies with others.

To address the above challenges, we propose a new federated A2C framework called "Advantage Actor Federated Critic (A2FC)". Unlike a traditional application of FL in the MARL system where a central server averages the weights of the agents' actor and critic models separately and sends the average weights back to the agents, the proposed framework only aggregates the weights of agents' critic models. Our proposed framework is especially suited to MARL systems where agents have heterogeneous action spaces. Also, the framework does not require any communication among agents, which significantly reduces the communication overhead as well as protects the privacy of agents' policies in MARL systems.

The main contributions of this paper can be summarized as follows: we propose a novel federated A2C algorithm for MARL systems with agents having heterogeneous action spaces. The proposed method averages the agents' critic neural network models only. It has the added benefit of 1) the personalization of agents' optimal actions; 2) reducing the communication overhead as agents in the MARL system do not need to communicate with each other; 3) privacy preservation as agents do not share their trained policies with others. We demonstrate the effectiveness of our proposed method in a comprehensive experiment in an ATSC system.

## 2 PRELIMINARIES AND CHALLENGES

### 2.1 POLICY-BASED RL ALGORITHMS

The Policy Gradient method, also known as REINFORCE, enhances a parameterized model $\pi_\theta$ by employing gradient descent to maximize the cumulative reward over the long term. The parameters $\theta$ are iteratively adjusted to increase the likelihood of selecting a sequence of optimal actions, utilizing sampled total rewards. Assuming a trajectory $(s_1, a_1, r_1, s_2, a_2, r_2, \cdots, r_T)$ on the parameterized model $\pi_\theta$, the loss function is formulated as follows:

$$\mathcal{L}(\theta) = \hat{R}_t \sum_{t=1}^{T} \log \pi_\theta(a_t|s_t) \tag{1}$$

Here, the estimated total reward is denoted as $\hat{R}_T = \sum_{t=1}^{T} \gamma^t r_t$.

The A2C algorithm integrates the characteristics of Q-learning into the features of the policy gradient method. It introduces a value function $V_\omega$ to estimate the expected reward $\mathbb{E}_\pi[R_t^T|s_t = s]$. The estimated total reward is revised to $\tilde{R}_T = \hat{R}_T + \gamma^t V_\omega(s_T)$. The algorithm defines the Advantage function as $A_t = \tilde{R}_t - V_\omega(s_t)$, which is utilized to update the neural network parameters of both

the Actor ($\theta$) and the Critic ($\omega$). The Advantage function effectively assesses the performance of the agent's action in the current step by comparing it to other possible actions.

## 2.2 MULTI-AGENT REINFORCEMENT LEARNING

MARL focuses on studying the behavior of multiple agents within a shared environment, where all agents collectively observe the environment's state and receive rewards based on their joint actions. Let's consider a network $G(\mathcal{V}, \mathcal{E})$ that represents interactions among multiple agents in a MARL system. Each agent $i \in \mathcal{V}$ can take discrete actions $a_i \in \mathcal{A}_i$, communicate with a neighbor represented by edge $i, j \in \mathcal{E}$, and share a global reward $r(s, a)$. Many MARL algorithms are grounded in the context of Q learning. These algorithms allow agents to observe the global state and collaboratively take actions that contribute to a decomposable global Q function $Q(s, a) = \sum_{i \in \mathcal{V}} Q_i(s_i, a_i)$. However, merely combining MARL with Q learning disregards agent interactions, resulting in worse convergence at the global state. Coordinated Q learning strikes a balance between optimality and scalability through iterative message passing and control synchronization among neighboring agents. In this approach, $Q_i(s, a) \approx \sum_{j \in \mathcal{N}i} Mj(s, a_j, a_{\mathcal{N}j})$, where $\mathcal{N}i$ refers to the set of neighbors of agent $i$, and $M_j$ represents the message from neighbor $j$ of agent $i$. Independent Q learning is a fully scalable strategy that omits message passing. In this case, all local Q functions are dependent solely on the local action: $Q_i(s, a) \approx Q_i(s, a_i)$. Each agent must possess knowledge of other agents' policies and implicitly consider their behavior as part of the environment dynamics in continuous policy training.

Q learning is inherently reliant on tabular structures, which limits the number of states and actions due to real-world device storage and computation constraints. Independent Advantage Actor-Critic (A2C) extends the principles of independent Q learning to the actor-critic framework where agents still share the global state and reward. Each agent individually trains its policy $\pi_{\theta_i}$ and the corresponding value function $V_{\omega_i}$, estimating the local return as $R_{t,i} = \hat{R}t + \gamma^t V\omega_i(s_T|\pi_{\theta_i})$. Federated A2C further extends Independent A2C from the multi-agent environment to the FL scenario using the FedAvg algorithm. In this approach, each agent updates its policy $\theta_i$ and value function $\omega_i$ parameters locally, which are then aggregated on a centralized server. The server computes the mean parameters, i.e., $\theta = \frac{1}{|\mathcal{V}|} \sum_{i \in \mathcal{V}} \theta_i$ and $\omega = \frac{1}{|\mathcal{V}|} \sum_{i \in \mathcal{V}} \omega_i$, and sends them back to the agents for training in the next round.

## 2.3 CHALLENGES

While MARL algorithms have shown significant progress over the past decade and have found widespread practical applications, they still face several challenges and limitations. One of the key issues is that existing MARL algorithms often rely on partial observations of the system through communication with neighboring agents. This can lead to slow convergence and hinder the system from reaching global optimality through joint actions of the agents. As each agent's policy might only converge locally, the global optimality is not guaranteed. Moreover, the communication among agents can cause high latency and communication overhead, rendering these algorithms less suitable for applications requiring low latency. Additionally, the assumption that agents can freely communicate and share states and trained policies doesn't always hold in practice. Agents might only be able to train their policies independently based on their own observations, making the assumption of full communication unrealistic. The sharing of policies also raises privacy concerns, particularly in scenarios where sensitive data is involved. This makes it challenging to implement MARL in contexts where data privacy is a priority. While the FL framework offers a solution by allowing agents to upload model parameters for aggregation on a central server without requiring direct communication, the current federated A2C approach necessitates the transmission of both policy and value function parameters. This introduces significant communication overhead and latency, which could be impractical in applications where real-time decision-making is crucial. In summary, while MARL algorithms have made strides, these challenges of partial observation, slow convergence, communication overhead, privacy concerns, and latency need to be addressed for their broader and more effective application in various domains.

## 3 ADVANTAGE ACTOR FEDERATED CRITIC (A2FC)

### 3.1 OVERVIEW OF METHOD

To address the aforementioned challenges, we propose a novel framework called Advantage Actor Federated Critic (A2FC). In this framework, agents are required to upload only the parameters of their value functions (critic models) to a central server for aggregation, rather than transmitting both policy model (actor model) and value function parameters. The schematic overview of our proposed framework is illustrated in Figure 1.

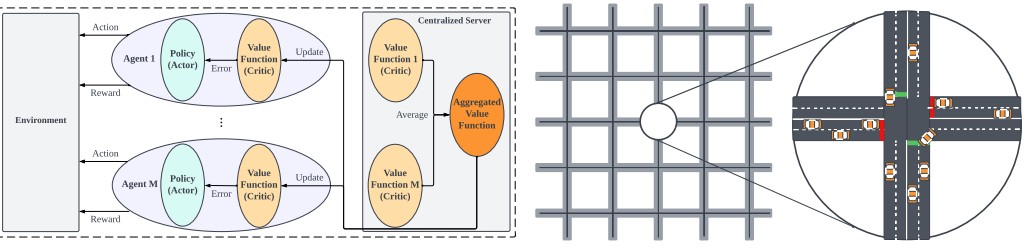

Figure 1: Overview of A2FC                    Figure 2: A $5 \times 5$ traffic grid

Consider a scenario with $M$ agents working collaboratively within an environment. At each step, every agent observes the current state of the environment and selects an action that advances the system to the subsequent state. The environment subsequently provides each agent with a reward based on its chosen action, which is then used to update both the agent's critic and actor models. The critic model approximates the expected reward using the obtained reward and subsequently guides the actor model towards making optimal decisions. Once agents have iteratively updated their models for a predetermined number of iterations, they transmit their value function parameters to a central server designated for aggregation. At the central server, the parameters of the agents' critic models are averaged, and the resulting average parameters are sent back to all agents and then used to initialize the next training iteration. This iterative process continues, thereby enabling the collaborative refinement of the agents' critic models within the A2FC framework.

Our proposed approach offers several key advantages over previous MARL algorithms. First, unlike traditional federated A2C methods, our approach accommodates agents within MARL systems that possess heterogeneous action spaces. This flexibility allows agents to have distinct action options, which is a departure from the conventional assumption of uniform action spaces. Secondly, our method streamlines communication, necessitating agents to interact solely with the centralized server, as is characteristic of FL techniques. There is no requirement for agents to communicate with one another or possess knowledge about the system's topology or neighborhood. This simplifies model optimization processes and reduces overall complexity. Thirdly, the approach also minimizes communication overhead significantly. Agents are solely tasked with uploading their value function parameters to the server and retrieving the averaged parameters subsequently. This streamlined communication process ensures efficient information exchange, improving system efficiency. Our approach also aligns with the privacy-preserving principles of the FL framework Shen et al. (2022). Agents retain their individual policies and do not communicate these policies with other agents. This design ensures privacy and safeguards sensitive information.

### 3.2 MULTI-AGENT ENVIRONMENT FORMULATION

To provide a comprehensive understanding of our proposed approach, we initially establish the framework of a MARL system and introduce key notations. In this context, consider a network denoted as $G(\mathcal{V})$, which embodies a MARL system featuring a set of $|\mathcal{V}|$ agents. Within this network, each individual agent $i \in \mathcal{V}$ engages in individual RL task, characterized by an underlying MDP tuple $\langle \mathcal{S}_i, \mathcal{A}_i, \mathcal{T}_i, \mathcal{R}_i \rangle$. Within this MARL system, each agent's observation capabilities are confined to its localized environment, encompassing only a portion of the complete environment. While agents' regions of influence might overlap with one another, their combined coverage must encompass the entire system's environment. Throughout each time step, all agents collaboratively explore their

respective accessible environments and concurrently make decisions. Notably, an individual agent's choices can impact the observable environment and conditions for other agents. During exploration, agent $i$ obtains a feedback reward $r_i$ from the environment, which serves as a feedback of this action. This exploration experience is then logged as a tuple within the agent's memory.

In this defined MARL system, the agents employ A2C algorithms to accomplish their respective RL tasks. Each agent $i$ owns an actor model neural network governed by parameters $\theta_i$ and a value function neural network governed by parameters $\omega_i$. The role of value functions is twofold: they enhance agents' comprehension of the environment dynamics and reward structure, while also assisting actor models in optimal action selection. Thus, each agent's objective revolves around updating both the actor and value function neural networks with information learned from their exploration experiences. This adaptation process aims to facilitate the acquisition of optimal policies $\pi_{\theta_i}$, resulting in the selection of actions that align with the encountered environmental states or their analogs.

### 3.3 PERSPECTIVE OF SINGLE AGENT

In contrast to previous MARL systems that assume sharing of global rewards and states among agents, we adopt a more flexible approach to address communication overhead and privacy concerns in specific applications. In our approach, agents can access their local rewards derived from the actions they execute. This approach eliminates the need for sharing a global reward, while maintaining a global reward function. Specifically, agents estimate local returns using the following expression:

$$\tilde{R_{t,i}} = \hat{R_{t,i}} + \gamma^t V_{\omega_i}(s_T | \pi_{\theta_i}) \tag{2}$$

Importantly, the gradient of the value function $\Delta\mathcal{L}(\omega_i)$ remains consistent for each agent. This consistency arises from the fact that $\hat{R_{t,i}}$ is sampled using the same stationary policy $\pi_{\theta_i^-}$ for each agent $i$. The value function $V_{\omega_i} : \mathcal{S} \times \mathcal{A}i \to \mathbb{R}$ is responsible for estimating the future return for the policy $\pi\theta_i$, and this estimation plays a crucial role in computing the gradient of the actor model $\Delta\mathcal{L}(\theta_i)$.

In the proposed MARL system, each agent's local policy and value function are solely dependent on the input state $s_t$. The loss function for the value function is formulated as follows:

$$\mathcal{L}(\omega_i) = \frac{1}{2|T|} \sum_{t=1}^{T} (\tilde{R_{t,i}} - V_{\omega_i}(s_{t,i}))^2. \tag{3}$$

Here, $|T|$ represents the total number of steps, $\tilde{R_{t,i}}$ is the locally estimated return for agent $i$ at time $t$, and $V_{\omega_i}(s_{t,i})$ stands for the value function's prediction for the state $s_{t,i}$.

The advantage value is defined as $A_{t,i} = R_{t,i} - V_{\omega_i^-}(s_t)$, and based on this definition, the loss function for the actor model is given by:

$$\mathcal{L}(\theta_i) = \frac{1}{2|T|} \sum_{t=1}^{T} \log \pi_{\theta_i}(a_{t,i} | s_{t,i}) \cdot A_{t,i}. \tag{4}$$

In this equation, $\pi_{\theta_i}(a_{t,i} | s_{t,i})$ represents the probability assigned by the actor model to the action $a_{t,i}$ in state $s_{t,i}$. The loss function seeks to maximize the probability-weighted advantage value, which essentially encourages the policy to prefer actions that lead to higher advantages.

However, the abilities of agents in this system are limited to observing only a portion of the environment. Consequently, they may struggle to collectively optimize global performance without a form of communication. This can lead to agents only achieving local optimization, rather than reaching the global optimum. To overcome this challenge and enable the merging of agents' local observations in a way that respects privacy, we propose using FL framework.

Traditional FL methods typically involve aggregating agents' actor models and value functions. However, this approach assumes that all agents have the the same set of actions, denoted as $|\mathcal{A}i| = |\mathcal{A}j|$ for any two agents $i$ and $j$. This requirement stems from the necessity for the agents'

neural network structures to match in order to aggregate weights effectively. In practical scenarios, such as in smart traffic control systems, agents may have differing action spaces due to individual variations—like traffic intersections with unique permissible signals. This diversity of action spaces poses a challenge to traditional FL methods in MARL systems.

## 3.4 A2FC ALGORITHM

The A2FC algorithm we propose brings forward a notable departure from the conventional methods. Unlike previous approaches, A2FC doesn't require agents to possess matching action spaces or an equal number of actions. Its innovation lies in the integration of the FedAvg algorithm, taken from FL, into MARL systems where we only aggregate the weights of agents' critic models instead of both actor and critic models. This unique design allows agents to independently train their actor models, enabling personalized action improvement within their distinct exploration domains. Simultaneously, the aggregation of critic models tackles the issue stemming from agents having access to partial observations. The averaged critic model weights encapsulate all agents' experienced states and their understanding of the environment's reward setting. Algorithm 1 provides a concise overview of our proposed A2FC framework within the context of a MARL system.

---

**Algorithm 1** A2FC Algorithm

---

**Require:** $T, E, \eta_\omega, \eta_\theta$

1: **Initialize** $s_0 = \{s_{01}, ..., s_{0i}\}$, $B = \{B_1 = \phi, ..., B_i = \phi\}$, $\pi_0 = \{\pi_{01}, ..., \pi_{0i}\}$, $t \leftarrow 0, e \leftarrow 0,$
2: **repeat**
3:     $t \leftarrow t + 1$
4:     **for** $i \in \mathcal{V}$ **do**
5:        Select action $a_{t,i} \leftarrow \pi_{t,i}$; Obtained reward $r_{t,i} \leftarrow \mathcal{R}_i(s, a_{t,i})$
6:        The next state $s'_{t,i} \leftarrow \mathcal{T}_i$; $B_i \leftarrow B_i \cup \{(t, s_{t,i}, \pi_{t,i}, a_{t,i}, r_{t,i}, s'_{t,i})\}$
7:     **end for**
8:     **if** $t = T$ **then**
9:        **for** $i \in \mathcal{V}$ **do**
10:           Estimate $\tilde{R}_{\tau,i}, \hat{R}_{\tau,i}, \forall \tau \in T$; Update $\omega_i$ with $\eta_\omega \nabla \mathcal{L}(\omega_i)$; Update $\theta_i$ with $\eta_\theta \nabla \mathcal{L}(\theta_i)$
11:           $s_i \leftarrow s_{0i}, \pi_i \leftarrow \pi_{0i}, t \leftarrow 0$
12:        **end for**
13:        $e \leftarrow e + 1$
14:     **end if**
15:     **if** $e = E$ **then**
16:        $\omega = \frac{1}{|\mathcal{V}|} \sum_{i \in \mathcal{V}} \omega_i$; $\omega_i \leftarrow \omega, \forall i \in \mathcal{V}$
17:        $e \leftarrow 0$
18:     **end if**
19: **until** Training Ends

---

## 3.5 METHOD DISCUSSION

The critic model plays a crucial role in helping agents comprehend the reward structure of the environment. By taking an agent's current state or observations as input, the critic model predicts an expected reward achievable through the agent's chosen action. The primary objective of a critic model is to maximize this expected reward within the given state. In a MARL environment, the reward function remains consistent across all agents, even though their exploration is limited to certain parts of the environment. Agents' training experiences can contribute to each other's understanding of the global environment. The FedAvg algorithm proves to be a suitable approach for enhancing agents' awareness in a MARL system. It accomplishes this by averaging the critic model weights from all agents. This aggregation incorporates agents' past training experiences and their collective comprehension of the reward mechanism from a diverse range of visited states. When the centralized server provides agents with the averaged model, they gain insights into the global environment through others' experiences. This averaged critic model offers improved generalization for agents, especially when encountering unfamiliar states that other agents have previously encountered. As a

result, the loss function in Eq. 3 transforms to:

$$\mathcal{L}(\omega_i) = \frac{1}{2|T|} \sum_{t=1}^{T} (\tilde{R_{t,i}} - V_{\omega(s_{t,i})})^2, \tag{5}$$

where $\omega = \frac{1}{|\mathcal{V}|} \sum_{i \in \mathcal{V}} \omega_i$. This average critic model enhances agents' ability to cope with new scenarios, thus contributing to more effective learning in a cooperative MARL system.

In contrast, agents' actor models need to retain a more personalized character in order to determine their optimal policies. The actor model operates by taking the agent's current state or observations as inputs and generating a probability distribution across the agent's action space. Given that agents typically possess distinct action spaces, traditional MARL methods that employ FL cannot directly aggregate agents' actor models with varying structures. Furthermore, it is crucial to preserve the personality of agents' actor models, as this enables them to take optimal actions within their specific environment. A straightforward aggregation of agents' actor models by assuming a shared action space among all agents would lead to reduced model personalization and consequently result in worse performance. Therefore, addressing the personalized nature of actor models is essential for successful and effective learning in a cooperative MARL system.

From a privacy preservation perspective, our proposed method ensures that agents' trained policies remain private. Since our approach doesn't necessitate direct communication among agents, our method enhances the security of agents' actor models by enabling them to be trained individually without any policy sharing. We consider that an agent's actor model contains a more substantial portion of private information compared to their critic model. An actor model updates based on the agent's exploration experiences, encompassing tuples involving visited states, chosen actions, acquired rewards, and ensuing states stored in the agent's memory. Consequently, the actor model directly exposes the agent's learned policy, raising privacy concerns. On the contrary, while a critic model also depends on agents' states or observations as inputs, its focus is centered on estimating the broader environmental reward structure, which is publicly available to all agents. Therefore, the aggregation of critic models doesn't result in any private information leakage from the agents. By design, our approach mitigates privacy risks related to policy exposure, making it a promising solution for secure MARL systems.

## 4 EXPERIMENTS

This section outlines the experimental implementation of the A2FC algorithm within an ATSC environment, utilizing the microscopic traffic simulator SUMO Krajzewicz et al. (2012). Our experimental setup draws inspiration from a pre-existing study Chu et al. (2019). For evaluation purposes, we contrast our proposed approach against two cutting-edge methods in the field of ATSC, namely Multi-Agent Advantage Actor-Critic (MA2C) and Independent Advantage Actor-Critic (IA2C). These benchmarks provide a comprehensive comparison to gauge the effectiveness of our A2FC algorithm within the ATSC context.

### 4.1 TRAFFIC GRID AND PARAMETER SETUPS

The experiment involves an MARL framework applied to an ATSC scenario within a SUMO-simulated Krajzewicz et al. (2012) traffic environment from a pre-existing study Chu et al. (2019). The traffic environment is designed as a 5x5 synthetic traffic grid, illustrated in Figure 2. In this traffic grid, two-lane arterial streets have a speed limit of 20 m/s, while one-lane avenues have a speed limit of 11 m/s. Each intersection within this grid comprises five possible phases, including East-West straight, East-West left-turn, and three phases for straights and left-turns for East, West, and North-South directions. Additionally, the grid incorporates four time-variant traffic flow groups, accommodating varying levels of traffic demands.

Traffic flows in real-world scenarios are characterized by their complexity in both spatial and temporal dimensions. To capture temporal dependencies in our scenario, we employ a Long Short-Term Memory layer as the final hidden layer of the DNN. This is the most straightforward method to involve feeding all historical states into an A2C agent. Notably, our proposed method focuses on aggregating agents' critic models, and thus, we train actor and critic DNN models separately. The

experiment comprises 1 million training steps, each with a duration of 720 steps, resulting in approximately 1400 episodes. In our MDP, we set the discount factor $\gamma$ to 0.99 and the exploration factor $\alpha$ to 0.75. The learning rates used in Algorithm 1 are configured as $\eta_\theta = 5e - 4$ for actor models and $\eta_\omega = 2.5e - 4$ for critic models.

## 4.2 EVALUATION METRICS

In our experiment, we conducted a comparative analysis between A2FC, MA2C, and IA2C, which are all developed based on the application of the A2C algorithm in MARL systems, particularly for ATSC. We evaluated these methods by measuring their training rewards throughout the training process. Additionally, we assessed the trained policies in terms of several key performance metrics, including average queue length, average intersection delay, and average vehicle speed within the traffic grid. Ideally, a well-trained policy should exhibit short and stable queue lengths, minimal intersection delays, and a consistent, higher vehicle speed.

## 4.3 EXPERIMENTAL RESULTS

Figure 3 displays the curves representing the average training rewards of A2FC, MA2C, and IA2C throughout the training process. These curves depict the increasing rewards that gradually converge as the agents accumulate experience and refine their local optimal policies. The data presented in this figure demonstrate that A2FC exhibits strong performance during the training process. While it may have a slightly slower convergence rate compared to MA2C, mainly because the centralized server aggregates agents' critic models only every 720 steps, A2FC ultimately converges to a higher reward, approximately -490. This convergence level is similar to MA2C's convergent reward of around -500 but significantly better than IA2C. Our method achieves a more stable convergence of rewards at the end of the training stage compared to MA2C's larger fluctuations.

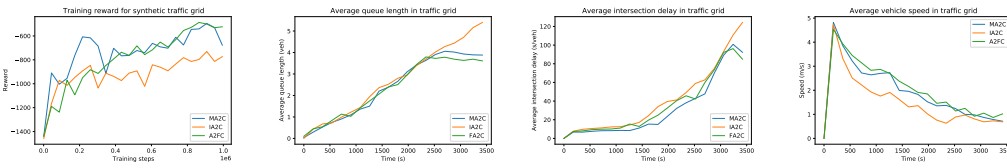

Figure 3: Training reward in traffic grid

Figure 4: Average queue length in traffic grid

Figure 5: Average intersection delay in traffic grid

Figure 6: Average vehicle speed in traffic grid

Our method outperforms the others primarily due to its ability to provide a global view of the environment through the aggregation of agents' critic models. As previously explained, an agent's critic model encapsulates its comprehension of the reward function in the MARL system and its past environmental observations. The aggregation of these critic models imparts a comprehensive perspective of the environment and the reward function to all agents. Consequently, agents can achieve convergence on a global scale.

In contrast, IA2C struggles to converge globally due to its limited communication, which is confined to a small region. MA2C enhances agents' environmental awareness by sharing more information about neighboring policies, but it still falls short of granting access to a complete global view. This limitation contributes to the fluctuations observed in MA2C's convergent rewards, as mentioned earlier. Additionally, MA2C necessitates extensive policy sharing among agents at each training step, resulting in high communication overhead compared to A2FC. Therefore, our proposed method excels in the training process, achieving a stable convergence to a higher reward while minimizing communication overhead.

Figure 4 provides insight into the average queue length of all policies at each step in the experiment. Notably, IA2C exhibits a concerning trend with a continuously increasing average queue length, indicating its inability to effectively alleviate congestion in the system. Conversely, both A2FC and MA2C demonstrate more stable and sustainable policies. They achieve lower congestion levels and faster recovery. Figure 5 illustrates the average intersection delay of the three methods. This metric

follows a similar pattern to the queue length, with all methods experiencing an initial increase due to rising vehicle volume. However, in the later stages of testing, both A2FC and MA2C effectively decrease the average intersection delay, signifying their capacity to achieve traffic congestion recovery. In contrast, IA2C struggles to mitigate congestion throughout the testing period. Our proposed method exhibits an advantage by achieving stability earlier and reducing the average intersection delay to a slightly lower level. This demonstrates the efficacy of A2FC in handling traffic congestion scenarios.

Figure 6 showcases the average vehicle speed achieved by the three trained policies. A2FC exhibits the highest average vehicle speed among the three policies. Despite a general decrease in average speed across all policies as vehicle volume increases, A2FC and MA2C demonstrate more stable performance and converge to a slightly higher average speed. This suggests that these policies are effective in reducing traffic congestion and improving vehicle flow. In contrast, IA2C lags behind, displaying the poorest performance of the three methods. This is primarily attributed to its slower convergence, resulting in a dramatic decrease in average vehicle speed and increased traffic congestion. Table 1 provides a summary of the experimental results, offering a comprehensive overview of the metrics for all three methods. It reinforces the superior performance of A2FC in mitigating congestion and enhancing traffic flow.

| Metrics | Temporal Averages | | |
|---|---|---|---|
| | MA2C | IA2C | **A2FC** |
| Avg. training reward | -738.3 | -921.5 | **-794.5** |
| Con. training reward | -499.4 | -808.1 | **-492.8** |
| Avg. queue length (veh) | 2.4 | 2.7 | **2.3** |
| Avg. intersection delay (s/veh) | 31.7 | 40.4 | **34.8** |
| Avg. vehicle speed (m/s) | 1.9 | 1.5 | **2.1** |

Table 1: ATSC performance of all metrics in the traffic grid

## 5 RELATED WORK

Federated RL (FRL) can be dividied into Horizontal (HFRL) and Vertical (VFRL) depending on environment partition (Qi et al., 2021). Numerous scholarly contributions have introduced methodologies within HFRL, characterized by agents' environments operating independently, albeit with aligned state and action spaces. Liu et al. (2019) employed a knowledge fusion algorithm based on generative networks, whereas in the present study, a federation policy dictates the approach for model fusion. Wang et al. (2020b) presented a federated DRL-based edge caching framework for IoT networks, aiming to reduce redundant traffic and enhance QoS. Lim et al. (2020) introduced a FRL scheme to control multiple real IoT devices of the same type but with slightly different dynamics. Unlike HFRL, several agents engage with a shared global environment in VFRL, each having access to a restricted subset of state information within their field of observation. VFRL presents a more intricate and less explored challenge in the current research landscape. Zhuo et al. (2019) considered both client models and the global model as components of a unified Q network, which is subsequently optimized utilizing the Bellman equation. Wang et al. (2020a) combined FL and A2C in ATSC systems. However, the method simply averages both critic and actor networks, which results in reduced personalization and lower global performance.

## 6 CONCLUSION

In summary, we have introduced a novel federated A2C framework known as A2FC, designed to address the challenges posed by agents with heterogeneous action spaces in MARL systems. A key feature of our approach is the utilization of a centralized server within the federated learning framework, which only aggregates agents' critic models, eliminating the need for communication among agents to share their policies. A2FC is well-suited for MARL systems where agents have diverse action spaces, offering substantial reductions in communication overhead as agents only exchange parameters related to critic models. Our method also prioritizes the privacy of agents during the training process. Our experiments conducted in an ATSC environment have demonstrated the effectiveness of our proposed method compared to other methods in MARL systems.

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
