# OpenReview forum: "A2FC: A FEDERATED ADVANTAGE ACTOR-CRITIC LEARNING APPROACH FOR HETEROGENEOUS ACTION SPACES"
_ICLR.cc/2024/Conference — ICLR 2024 Conference Withdrawn Submission_

### Official Review · Reviewer_Bftb · 2023-10-23

**Soundness:** 2 fair
**Presentation:** 3 good
**Contribution:** 2 fair
**Rating:** 3
**Confidence:** 3

**Summary:**

This paper proposes a federated learning method for multi-agent reinforcement learning (MARL). In the proposed Advantage Actor Federated Critic (A2FC) algorithm, the central server only aggregates the critic models of the agents, leaving the training of actor models to the individual agents’ local machines. The authors claim that the benefits of this method includes preserving the privacy, allowing personalized action spaces of the agents, and mitigating the communication overhead. The authors demonstrate the advantages of their method by simulating on an adaptive traffic signal control system.

**Strengths:**

This paper studies an interesting problem in multi-agent RL. The proposed algorithm is clean and easy to understand. The simulation results demonstrate the advantages of the proposed method over existing works.

**Weaknesses:**

While this paper delivers some interesting ideas, the significance and amount of technical contributions of the paper are not sufficient for an ICLR paper. The algorithmic idea is very simple and straightforward and fails to discuss many important practical considerations. For example, if the agents do not share their actor models, the benefits that federated learning can expedite training by aggregating the agents’ policies no longer exist. This paper also does not discuss synchronous vs. asynchronous parameter sharing and its related issues, which in my opinion are also important topics when designing federated learning algorithms.

The simulation results are rather small scale: The authors only conducted simulations on one simple traffic signal control and compared with two baselines. In general, I would expect to see more comprehensive evaluations on diverse MARL benchmarks and comparisons with popular state-of-the-art baseline algorithms. From the numerical results, the improvement over existing methods is also marginal. In addition, the related work section is also far from comprehensive, and no proper reference is given in the majority part of the paper.

**Questions:**

Please respond to my concerns in the Weaknesses section if possible.

---

### Official Review · Reviewer_egev · 2023-10-31

**Soundness:** 1 poor
**Presentation:** 2 fair
**Contribution:** 1 poor
**Rating:** 3
**Confidence:** 4

**Summary:**

The authors are proposing an adaptation of the multi-agent advantage actor critic A2C algorithm to the setup of the federated learning. In the proposed algorithm, the critic model is "streamlined", while the training of the actor models are uploaded to the agent's local machines.

The proposed algorithm is trained in a scenario of adaptive traffic signal control. The claims of the paper are that the agents "optimal actions are personalized", "the communication overhead is reduced as the agents in the MARL system do not need to communicate with each other" and "privacy preservation as agents do not share their trained policies with each other".

**Strengths:**

* The authors seem to have a good understanding of the mathematical formulation of MARL.

**Weaknesses:**

* The claims of the paper are unclear. What does it mean that the "optimal actions are personalized?". How do we measure personalization?
* What kind of communication overhead we are talking about, during training or during inference? How big is the communication cost for a stoplight? Although this is one of the three claims, the paper does not seem to measure communication cost anywhere in the rest of the paper.
* Why is this approach more privacy preserving than other federated learning approaches? Is privacy preservation an issue for traffic signal control, i.e. one traffic signal not to know what is the color of the next one? One would think that this is a very bad example of an application of federated learning.

**Questions:**

* Can you reformulate the claims of the paper in such a way that they (a)
show only what this paper does beyond the state of the art and (b) they are measurable or provable and supported by the experiments or theoretical work in the paper?

---

### Official Review · Reviewer_PA69 · 2023-11-09

**Soundness:** 1 poor
**Presentation:** 1 poor
**Contribution:** 1 poor
**Rating:** 3
**Confidence:** 3

**Summary:**

This paper proposes a method that combines federated learning (FL) and multiagent reinforcement learning (MARL) called A2FC. The motivation behind the method is to address the potential privacy issues with previous federated learning-based MARL methods where an agent must share their model information with other agents during training. At the same time, A2FC is also designed to enable federated learning in cases where agents in the multiagent system have different action spaces. A2FC achieves this by designing each agent to have its separate policy and critic network that are individually trained, while the critic network of different agents is also periodically aggregated to combine the knowledge and experience of different agents.

A2FC is finally tested in an environment that simulates a traffic signal control problem. A2FC's performance across different metrics is specifically compared against multiagent A2C and independent A2C. The authors argue that their experiments indicate their method produces superior performance against compared methods across the different evaluated metrics.

**Strengths:**

(Minor Strength - Novelty - Critic Aggregation) To the best of my limited knowledge on the subject, A2FC provides a novel approach for MARL. I specifically have not seen a method that aggregates individual critics by averaging out their parameters like in A2FC.

**Weaknesses:**

**(Major Weakness - Soundness - Mistaken claims about MARL methods)**

This research is motivated by incorrect claims regarding MARL algorithms (and especially MAA2C). In both Section 1 and Section 3.1, the authors **claimed that MARL methods require agents to either (i) share their policy parameters or (ii) communicate to deal with partial observability**. However, while they require agents to share observation and action information, Centralized Training Decentralized Execution MARL methods can operate during execution time without having to share policy parameters or communicate. At the same time, it is incorrect that existing MARL methods require agents to be homogenous by having the same action space. A MARL benchmarking work from Papoudakis et al. (2021) did an extensive study on the effects of training different policy networks for agents not sharing a common action space.

**(Major Weakness - Soundness - Lack of Demonstrations Regarding The Need for Federated Learning)**

Another issue is the simplicity of the ATSC environment which does not adequately show the need for federated learning. In environments based on simple simulations like ATSC, all agents' policies, the centralized critic, and the simulation should fit in a single machine. In that case, we can leverage CTDE methods that operate on observations and train all the methods in the same machine without having to concern ourselves with privacy issues. During execution, the decentralized methods should then operate without having the need for any communication or information sharing, enabling the trained policies to work in different machines.

I believe to better show the need for federated learning, the authors have to use environments where centralized training is not possible either because of (i) models or (ii) simulations with large sizes that cannot fit within a single machine. In these environments, training requires real communication between machines and federated learning should be crucial. However, the simple ATSC environment used in this work seems too simple to show this need for federated learning.

**(Major Weakness - Soundness - Missing Baseline)**

There are potential baselines that can also (i) minimize exchanged information between an actor & centralized server and (ii) deal with heterogeneous agents. For instance, MAA2C (Papoudakis et al., 2021) where agents have separate policy networks and a shared centralized critic saved in the centralized server could also achieve this goal. Without having to exchange the parameters of the actor and centralized critic network, an agent can still learn if it sends:

1. Its achieved rewards,
2. Its selected action,
3. Its observation,

to the centralized critic in the centralized server. In turn, the centralized critic can update itself and return an advantage function that each agent can use to update their respective policy networks.

Arguably, the above MAA2C method would be preferable since agents will not have to send neural network parameters, which (i) usually use more communication bandwidth compared to the agent's individual rewards, actions, or observations. At the same time, sending these rewards, actions, and observations could better preserve privacy by disclosing less information about each agent's decision-making process.

Since this modification of MAA2C is suitable for the problems addressed in this work, it should have been included as a baseline.

**(Minor Weakness - Soundness - Problem Formulation)**

The choice of using MDPs to model the interaction between agents in the MARL problem is also questionable. In the case of MARL where teammates continuously change their policy as a result of learning, an MDP's Markov assumption will not hold and MDPs will not be adequate models of agent interaction. A better choice would be to use Decentralized POMDPs  (Oliehoek et al., 2012) or Stochastic Games as the problem formulation.

**(Major Weakness - Soundness - Missing Citations on Existing Deep MARL Methods)**

The paper is also missing lots of citations to existing Deep MARL methods. I would highly recommend the authors at least read the methods compared in the benchmarking paper from Papoudakis et al. (2021). The authors should then see that a lot of CTDE methods (aside from MAA2C) can actually address the problem tackled by their work.

**(Major Weakness - Soundness - Environment Selection)**

The current version of the work only has experiments on the A2FC environment. Compared with existing MARL methods that use at least three different environments for evaluation, I believe limiting the experiments to the A2FC algorithm will weaken the potential significance of the results, even if it is inherently positive. Further experiments need to be done in other environments to better illustrate the generality of the method.

**(Major Weakness - Soundness - Significance Testing)**

There is also a severe lack of significance testing in the analysis section. The authors only provided average returns resulting from their method without demonstrating whether those differences are significant or not. I would refer the authors to the work of Agarwal et al. (2021) for example statistical tests when comparing different RL methods' performances.

**(Major Weakness - Clarity - Inadequate Description of Motivation Behind Research)**

Rather than explaining the differences between RL methods as in the first few paragraphs of the introduction, Section 1 could have been used to better motivate the importance of solving problems encountered in this work. For instance, this can be done by showing a diagram that outlines the exchanged information between agents and the central server. Furthermore, the authors could better argue why this information exchange is undesirable. There could also be some descriptions of why previous methods are not desirable or exhibit problems that the work wants to solve.

**(Major Weakness - Clarity - Description of MARL Problem being solved)**

The description of the type of MARL problems solved is quite confusing. Early in Section 1, it does seem that the paper addresses fully cooperative environments. Meanwhile, in the rest of the document, there are instances (for example in the problem formulation) where it seems agents can have differing reward functions. I would expect some consistency in detailing this important information.

Citations:
Papoudakis, Georgios, et al. "Benchmarking Multi-Agent Deep Reinforcement Learning Algorithms in Cooperative Tasks". NeurIPS 2021.
Oliehoek, Frans A. "Decentralized Pomdps." Reinforcement Learning: State-of-the-Art.
Agarwal, Rishabh, et al. "Deep reinforcement learning at the edge of the statistical precipice." NeurIPS 2021.

**Questions:**

1. Would CTDE methods work in solving the privacy concerns tackled by this paper?
2. Why use MDPs to formulate the decision-making problem?
3. How do you describe the relation between this work and other existing CTDE-based MARL methods?
4. Is the method limited to fully cooperative MARL environments?

**Details Of Ethics Concerns:**

I have no ethical concerns about this work.

---

### Meta-Review · Area_Chair_3P2t · 2023-12-03

**Metareview:**

The paper proposes a new federated learning technique in the context of multiagent RL.  However, the reviewers raised many concerns about the setup, motivation, clarity, related work and experiments.  This work is not ready for publication.

**Justification For Why Not Higher Score:**

The reviewers unanimously recommend rejection and there is no author rebuttal.

**Justification For Why Not Lower Score:**

N/A

---

### Decision · Program_Chairs · 2024-01-16

Reject